# Targeting Aging and Longevity with Exogenous Nucleotides (TALENTs): Rationale, Design, and Baseline Characteristics from a Randomized Controlled Trial in Older Adults

**DOI:** 10.3390/nu16091343

**Published:** 2024-04-29

**Authors:** Shuyue Wang, Lixia Song, Rui Fan, Qianqian Chen, Mei You, Meng Cai, Yuxiao Wu, Yong Li, Meihong Xu

**Affiliations:** 1Department of Nutrition and Food Hygiene, School of Public Health, Peking University, Beijing 100191, China; 2111210090@stu.pku.edu.cn (S.W.); 1710306221@pku.edu.cn (L.S.); fanruirf@bjmu.edu.cn (R.F.); 2211210120@stu.pku.edu.cn (Q.C.); 1610306135@pku.edu.cn (M.Y.); 1810306215@pku.edu.cn (M.C.); 2311210112@bjmu.edu.cn (Y.W.); liyong@bjmu.edu.cn (Y.L.); 2Beijing Key Laboratory of Toxicological Research and Risk Assessment for Food Safety, Peking University, Beijing 100191, China

**Keywords:** healthy aging, nucleotides, randomized trial, aging biomarkers, older adults

## Abstract

Nucleotides (NTs), important biomolecules involved in numerous cellular processes, have been proposed as potential candidates for anti-aging interventions. However, whether nucleotides can act as an anti-aging supplement in older adults remains unclear. TALENTs is a randomized, double-blinded, placebo-controlled trial that evaluates the efficacy and safety of NTs as an anti-aging supplement in older adults by exploring the effects of NTs on multiple dimensions of aging in a rigorous scientific setting. Eligible community-dwelling adults aged 60–70 years were randomly assigned equally to two groups: nucleotides intervention group and placebo control group. Comprehensive geriatric health assessments were performed at baseline, 2-months, and 4-months of the intervention. Biological specimens were collected and stored for age-related biomarker testing and multi-omics sequencing. The primary outcome was the change from baseline to 4 months on leukocyte telomere length and DNA methylation age. The secondary aims were the changes in possible mechanisms underlying aging processes (immunity, inflammatory profile, oxidative stress, gene stability, endocrine, metabolism, and cardiovascular function). Other outcomes were changes in physical function, body composition and geriatric health assessment (including sleep quality, cognitive function, fatigue, frailty, and psychology). In the RCT, 301 participants were assessed for eligibility and 122 were enrolled. Participants averaged 65.65 years of age, and were predominately female (67.21%). All baseline characteristics were well-balanced between groups, as expected due to randomization. The majority of participants were pre-frailty and had at least one chronic condition. The mean scores for physical activity, psychological, fatigue and quality of life were within the normal range. However, nearly half of the participants still had room for improvement in cognitive level and sleep quality. This TALENTs trial will represent one of the most comprehensive experimental clinical trials in which supplements are administered to elderly participants. The findings of this study will contribute to our understanding of the anti-aging effects of NTs and provide insights into their potential applications in geriatric healthcare.

## 1. Introduction

The World Demography Report shows that the world is experiencing an unprecedented rapid aging process, and all countries will face the economic, social, and political challenges, as well as other issues brought about by population aging. Aging in humans physically refers to a multidimensional process, in that all the changes are accumulated in a person over time [1,2]. These aging changes are responsible for the progressive increases in the chance of disease and death associated with them. The central hallmarks of age-related changes include genomic instability, telomere depletion, epigenetic changes, loss of protein homeostasis, deregulated nutrient sensing, mitochondrial dysfunction, cellular senescence, stem cell exhaustion, and altered intercellular communication [3]. The reported anti-aging star compounds/drugs that have been subject to preclinical and clinical trials targeting the above aging characteristics, achieving certain results [4,5,6,7,8], include rapamycin, anti-aging drugs (senolytics), metformin, acarbose, spermidine, NAD+ supplements, and lithium. However, the function of these compounds needs to be further confirmed, and there are still many adverse reactions reported. The intervention of food-derived nutritional active ingredients has the advantages of safety, multi-effect, economy and preventive long-term interventions, which is an important way to explore the intervention methods of aging.

Nucleotides (NTs), the building blocks of DNA and RNA, are the material basis of heredity, energy metabolism, and signal transduction, and play an important role in the growth and development of organisms, metabolism, reproduction, heredity, and aging [9,10]. In the field of food nutrition, nucleotides (NTs) are the main forms of nucleoside composition detection and functional ingredient application measurement, and nucleotides have important nutritional value for body health. As a food raw material approved by the State Food and Drug Administration, nucleotides have been widely used in infant formula milk powder/food, and health food/special medical use formula food [11,12,13,14]. Additionally, their safety was fully verified by a lifetime feeding experiment and four generation breeding experiment with high safety [15,16,17]. What is more, NTs have been found to have many biological functions, such as anti-aging [18], telomere length regulation [19,20], tumor inhibition [21], immune regulation [22,23,24], anti-inflammatory and antioxidant properties [25,26], learning and memory improvement [27,28], anti-fatigue [29], liver protection [30,31], intestinal protection [32,33,34], flora regulation [35,36], growth promotion [37,38], neuroprotective effect [39], and participation in one carbon unit metabolism.

The food sources of NTs are animal viscera (liver, kidney, brain, etc.) and seafood, lean meat, broth, fish, goose meat, yeast are also rich in NTs. Nucleic acids in food are mostly in the form of nuclear proteins. Nucleoproteins are broken down into nucleic acids and proteins by the action of gastric acid in the stomach. The nucleic acid is gradually hydrolyzed in the small intestine by various hydrolases in the pancreatic and intestinal fluids, and finally produce the bases and pentose [10]. Evidence suggests that NTs in the diet may be regarded as conditionally essential nutrients. Under normal circumstances, the body can synthesize nucleotides de novo and remediate them to maintain nucleotide levels in the tissue. However, when the body is in a special state (immunosuppression, recovery from injury, infection, specific disease states, and insufficient nutrient intake) or a special life stage (rapid growth and functional decline), the ability of NTs to synthesize from scratch is greatly reduced and the remedial capacity of nucleotide synthesis is insufficient or limited [22]. Therefore, obtaining NTs externally is a better mode of supplementation when the organism is under special circumstances, which plays an important role in saving energy consumption of de novo or remediation synthesis in the body, maintaining tissue nucleotide pool homeostasis, and improving and optimizing tissue function.

In the process of aging, the ability of NTs to synthesize from scratch is greatly reduced and the remedial capacity of nucleotide synthesis is insufficient or limited. Moreover, due to the increase of DNA repair and mitochondrial DNA synthesis and the depletion of metabolic enzymes, the content of deoxynucleotide triphosphates (dNTPs) decreases, leading to cell cycle arrest during aging. dNTPs supplementation can promote sustained cell proliferation [40]. NTs as synthetic raw material of dNTPs with potential anti-aging effects may have positive significance for delaying senescence. However, the comprehensive evaluation of exogenous nucleotides on the improvement of nutrition in the elderly population is still lacking. To address this knowledge gap, this randomized controlled trial aims to explore whether NTs have anti-aging effects and explore the possible pathways of the anti-aging effects of NTs. This manuscript describes the study protocol and baseline results for the TALENTs trial.

## 2. Methods

### 2.1. Trial Design

This was an exploratory study of participants from the TALENTs trial (Targeting Aging and Longevity with Exogenous Nucleotides), which was a pragmatic 4-month prospective single-center randomized control trial (Figure 1). Briefly, the TALENTs trial enrolled community-living people aged 60–70 years, to evaluate the efficacy and safety of nucleotides as an anti-aging supplement. Major exclusion criteria included those having terminal or serious illness, inability to communicate adequately with the researcher, and having already taken nucleotide supplements, etc. Following the provision of written informed consent, baseline assessments were conducted. The demographic factors, lifestyle, general clinical data, current medication or supplementation use, and diet of the subjects were registerd. Additionally, the subjects underwent comprehensive health assessments, including clinical health physical examination, questionnaire surveys, physical function assessment, and anthropometrics as part of the screening process. Eligible participants were randomly assigned by a 1:1 ratio to receive supplementation of nucleotides (1.2 g/day) or placebo daily for 4 months. Assessments and biological sample collection at baseline, 2 months, and 4 months post-intervention were executed by well-trained and skilled researchers.

### 2.2. Ethic Committee Review and Approval

The study design and procedures have been conducted under the guidance of the Declaration of Helsinki and later amendments. Study approval has obtained from the Biomedical Ethics Committee of Peking University (ethics review approval number: IRB00001052-21114, date of approval: 17 November 2021). The TALENTs trial has been registered prospectively at Clinical Trials.gov (NCT05243018). The purpose, nature, and potential risks of participation in the trial were fully explained to the participants, and all participants provided written informed consent before participating in the trial.

### 2.3. Sample Size Calculation

A power analysis was not conducted due to the exploratory and preliminary nature of this study. Therefore, the sample size was based on the requirements of the human health food test, with 50 people in each group and an estimated loss of follow-up rate of 20%, so 60 people were required in each group; a 1:1 control group and intervention group totalled of 120 people.

### 2.4. Recruitment and Eligibility Criteria

In this study, subjects were recruited by the project leader and researchers through the method of publishing recruitment inspiration and posters on the Internet. As the trial site was set in Chengdu, potential participants were required to be local community residents and non-hospital patients. Research staff were to fully explain the purpose, nature, and potential risks of participation in this trial for prospective participants before trial participation (consent shown in Appendix A). Then, participants who provided written informed consent were assessed for their eligibility by comprehensive geriatric health assessments. Clinical physical examinations were conducted at Aikang Guobin Medical Examination Hospital (Chengdu, China) by clinical doctors with medical qualifications. Researchers evaluated participants’ final eligibility based on the trial’s inclusion and exclusion criteria.

Inclusion criteria: (1) age between 60 and 70 (2) no serious physical or mental illness; (3) never take nucleotide-related supplements/health food (4)Be able to follow the test protocol and sign the informed consent.

Exclusion criteria: (1) patients with related confirmed diseases, such as autoimmune system diseases, serious cardiovascular and cerebrovascular diseases, major organ complications such as in the liver and kidney, or complicated with other serious diseases such as malignant tumors, pancreatic diseases and mental diseases; (2) having abnormal screening laboratory test values or other lab test results that would preclude study participation in the judgment of the investigator; (3) severe visual or hearing loss affecting communication; (4) being participants in other clinical trials within the last 6 months; (5) having used food or medicine that is relevant to the function being tested.

### 2.5. Interventions

The intervention group was to receive the capsules of exogenous NTs, which were extracted from cane sugar by enzymatic hydrolysis, provided by Zhen-Ao Biotechnology Co., Ltd., (Dalian, China). The placebo capsules in the control group looked and tasted as similar as possible to those in the intervention group, making the participants indistinguishable. Both groups needed to take four capsules daily. Among them, each capsule in the nucleotides group contained 0.1 g starch excipients + 0.3 g nucleotides, and each capsule in the control group contained 0.4 g starch excipients. The specific nucleotide composition in this trial was 5′-AMP:5′-CMP:5′-GMPNa2:5′-UMPNa_2_, prepared according to the ratio of 16:41:19:24, which is consistent with the ratio in breast milk and meets the requirements of national infant formula addition and special medical food formula. The dose of 1.2 g/day used is currently approved as the ingredient dose of nucleotides in conventional commercially available health food products. The experimental intervention continued for 4 months without interruption.

### 2.6. Randomization and Blinding

In this trial, participants were to be centrally randomized at a 1:1 allocation ratio into intervention and control groups using computer-generated random numbers. After grouping, the main factors that may affect the results and outcome indicators, such as gender and age, were tested for balance to ensure the comparability between groups. The random scheme was generated and known only by the experiment designer. Study participants and data collection staff including investigators, clinicians and follow-up staff were blinded. Pharmacists from Zhen-Ao Biotechnology Co., Ltd., (Dalian, China) manufactured and packed each subject’s capsule into an opaque medicine bottle and numbered it on the outside. In the end, only the participant’s name was marked on the bottle, and the bottle packaging and the capsules contained were identical in appearance. Blinding was fully explained when subjects sign the informed consent. If subjects did not accept the blinding method, they were not included in this study. The effectiveness of the study blinding was assessed with a short questionnaire completed by the study subjects when the trial was finished. The randomization code would not be broken without consent from the experiment designer under unusual circumstances, such as in the event of numerous serious adverse events before the end of the study. Neither the trial designer nor the pharmacist participated in the trial.

### 2.7. Follow-Up

After entering the follow-up period of intervention, trained follow-up staff were responsible for follow-ups and regular home visits to distribute capsules to reduce the burden on participants. In addition to handing out intervention capsules during each home visit to the subjects, the follow-up staff also presented practical daily activities to keep the participants upbeat. A total of 120 subjects were randomly divided into 12 follow-up groups with 10 people in each group and followed up by 12 trained follow-up staff respectively. Participants were required to report the time taken, and dosage of intervention capsules daily in WeChat groups and report special circumstances or adverse reactions. Researchers provided participants with an interventional follow-up manual, which explains the content and time points they needed to cooperate with during the study, as well as what they should do and who to contact if any new symptoms or side effects occurred. This study specifically hired a clinical doctor to provide relevant clinical consultation for participants to understand their participation and participate in clinical management when necessary.

### 2.8. Outcomes

The primary outcomes assessed changes in molecular biomarkers associated with aging, encompassing telomere length and DNA-methylation clock compared with baseline. Secondary outcomes included alterations in mechanisms underlying aging processes such as immune function, inflammatory cytokine levels, oxidative stress levels, DNA stability, endocrine, metabolic, and cardiovascular functions. Other outcomes comprised physical function, body composition and the Geriatric Assessment Questionnaire (including sleep quality, cognitive function, fatigue, frailty, and psychology). Figure 2 lists the outcome measures. Data were obtained at baseline, 2 months, and 4 months by teams of fieldworkers, who were trained in standard measurement and protocols of questionnaire administration.

#### 2.8.1. Primary Outcomes

(1)Leukocyte telomere length: telomeres are repetitive nucleotide elements at the end of chromosomes that protect chromosomes from degradation and genetic information loss. In this study, leukocyte telomere length was measured using quantitative polymerase chain reaction assay and reported as T/S ratios (the relative ratio of telomere repeat copy number to single-copy gene) [41].(2)DNAm clocks: the DNAm clock, an algorithm that combines DNAm measurement information in the genome to quantify biological age changes, is considered a highly accurate molecular correlation of the true age of humans and can detect DNA methylation age reversal up to three months ahead of the respective individual clocks, so it has the potential to quantify biological aging and test life expectancy or assess intervention effectiveness [42,43,44].

#### 2.8.2. Secondary Outcomes

(1)Immune function: general parameters of immune health that are known to change with age were assessed as described, including IgA, IgG, IgM, IgE, and PBMCs (peripheral blood mononuclear cells) analyzed by FACS to assess the proportion and number of T lymphocyte subtypes (CD3^+^, CD3^+^CD4^+^, CD3^+^CD8^+^, CD4^+^CD25^+^);(2)Serum cytokine levels: SASP is a consequence of cell senescence and may occur in cells that, though undergoing cell cycle arrest, are still metabolically active and secrete proteins [45,46]. Luminex-based assays were be used, as these allow simultaneous measurement of different cytokines (TNF-α, IL-1β, IL-6, IL-18, TNF RI, TNF RII, ICAM-1, etc.);(3)Oxidative stress: enzyme-linked immunosorbent assay (ELISA) kits were used to detect the level of serum oxidative (MDA, NOX, SOD, GSH-Px, etc.);(4)Gene stability: γ-H2A.X phosphorylation is an initial and essential component of DNA damage foci and therefore a reliable marker of the extent of DNA damage [47,48];(5)Glycolipid metabolic profile: glucose, insulin, total cholesterol, HDL-C, LDL-C, triglycerides, etc.;(6)Endocrine and cardiovascular function: adiponectin, leptin, NT-proBNP, IGF-1, NO, carotid thickness of intima-media, spontaneous fluorescence of subcutaneous AGEs. etc.

#### 2.8.3. Other Outcomes

##### Geriatric Assessment Questionnaire (CGAQ)

A self-designed questionnaire was used to evaluate the health status of all subjects before and after intervention (Full Questionnaire shown in Appendix A). The specific contents of the questionnaire are as follows:(1)General information survey: demographic characteristics, including date of birth, occupation, education level, economic status, family status, etc.; lifestyle and living habits (smoking, drinking, tea, sports, etc.); the number of available teeth (including dentures) and the loss of teeth in the past one year; medical history (hypertension, coronary heart disease, obesity and other diseases); health food consumption, drug application history; expenditure on health, including physical examination, medicine, hospital expenses, etc.;(2)Quality of life: assessment of physical and mental functioning (SF-12) [49];(3)Frailty questionnaire: 28-item frailty index [50];(4)Cognitive status: Montreal Cognitive Assessment Scale (MoCA) [51];(5)Physical activity: Physical Activity Scale for the Elderly (PASE) [52];(6)Sleep quality: Pittsburgh Sleep Index [53];(7)Psychological function: Kessler Psychological Distress (K10) Scale [54] and Fatigue Scale-14 (FS-14) [55];(8)Dietary survey: dietary data were collected at baseline and after intervention using a 3 day, 24 h photo-assisted dietary intake assessment method to assess dietary stability during the trial [56]. The Food Frequency Questionnaire (FFQ) was used to measure the intake of foods rich in nucleotides, including seasonings high in nucleotides, to assess daily dietary nucleotide intake.

##### Anthropometry and Physical Function

Waist circumference, hip circumference, calf circumference, mid-arm circumference and neck circumference were measured with a tape measure to the nearest 0.1 cm. We used bioelectrical impedance analysis (BIA) to measure the proportion of whole-body lean and fat mass, body fat, trunk fat, body muscle, trunk muscle, and other components in the human body.

The level of physical functioning was assessed with the short physical performance battery (SPPB), which consisted of standing balance tasks (side-by-side, semi-, and full-tandem stands for 10 s each), a 4 m walk to assess usual gait speed, and time to complete five repeated chair stands. Handgrip strength and a 30 s sarm curl test were to measure upper muscle strength. The Tinetti clinical scale including balance assessment items and gait assessment items, was used to evaluate fall risk.

##### Clinical Health Physical Examination and Safety

All blood measurements were performed in the morning after an overnight fast. Blood, urine, stool routine examination, and liver and kidney function were examined, and other safety indicators included chest X-ray, lung function, and electrocardiogram, etc.

The number and types of adverse events including changes in laboratory parameters (blood chemistry, lipid profile), monitored and documented. All adverse events, including falls, fractures, hospitalizations, and deaths were recorded in an “adverse events diary” during follow-up visits by the trained follow-up staff, and by self-report during the study period.

##### Multi-Omics Sequencing

The biospecimens of the TALENTs trial (shown in Appendix A) were transferred to the laboratory on dry ice within 24 h of collection and then stored at −80 °C. Blood samples were collected in three different types of collection tubes containing distinct stabilizers or coagulants, including ethylenediaminetetraacetic acid (EDTA) tubes, serum separation tubes (SST), and RNA preservation tubes. Whole blood, blood clots, and serum were taken for whole genome methylation, metabolomic and transcriptomic sequencing. Stool samples were taken for metagenomic sequencing.

**Figure 2 nutrients-16-01343-f002:**
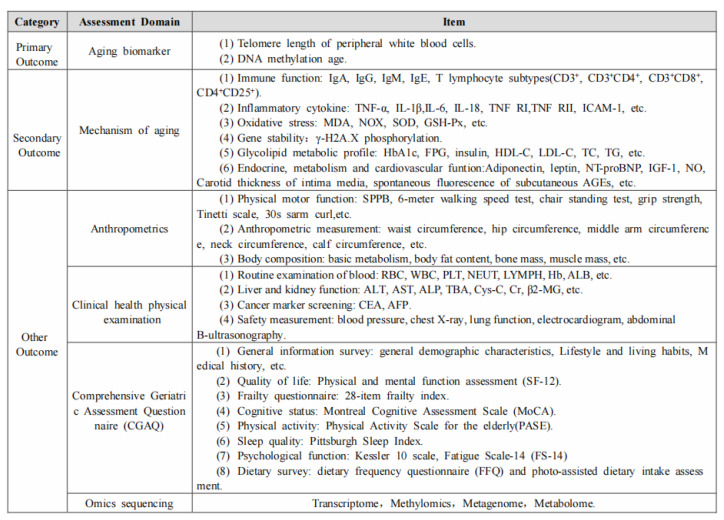
Assessment outcomes of the TALENTs trial.

### 2.9. Statistical Analysis

The data in this study were analyzed by intention-to-treat (ITT), and no interpolation was performed for data missing from general data. The Shapiro-Wilk test was used for data normality and the Levene test for variance homogeneity. If the data conformed to normal distribution, the measurement data are expressed as mean ± SD, and the measurement data of non-normal distribution are expressed as median (P25, P75).

In the inter-group comparison, the changes between the outcomes after intervention and the baseline data were used as the outcome variable, and independent sample T test was used for the normal distribution data. Non-parametric Mann-Whitney U test was used to compare the non-normal distribution measurement data groups. Generalized estimating equations (Generalized Estimating Equation, GEE) were used to assess the effect of nucleotide intervention of outcome indicators. GEE can tolerate the presence of missing values in longitudinal data, and can perform unbiased estimation of research information, and GEE is relatively stable in various job-related matrices, so it is widely recommended in intervention studies. There are two levels of longitudinal data in this study; the assessment at each time point is level 1, nested in the study object (i.e., level 2). Continuous outcome indicators at each time point were used as dependent variables to evaluate the group main effect, the time main effect and the interaction effect of the time group. In addition, GEE model was used to adjust covariates including age, gender, BMI, and the baseline of the variables corresponding to the results. Cohens D was calculated using the mean difference between the two groups, with d = 0.2 as a small effect size, d = 0.5 as a medium effect size, and d = 0.8 as a large effect size. Analyses were performed using the SPSS 26.0 software and overseen by an experienced biostatistician; *p* < 0.05 indicates a statistically significant difference.

### 2.10. Procedures

Figure 3 shows the timeline for scheduled assessments of the TALENTs trial.
(1)Recruitment and screening period: physical examination in the partner hospital (blood and stool samples retained at the same time), questionnaire survey, and other items were completed. Questionnaires were completed by trained investigators and participants face-to-face. At the same time, participants were required to carry their past medical records and current medication and health supplements for on-site registration. According to the physical examination results, the subjects unfit to be enrolled were excluded;(2)Initiation and follow-up period: during the intervention period, follow-up staffs are responsible for dispensing, reviewing, and documenting medications. Intervention capsules were distributed biweekly by follow-up staff during home visits. Old packaging products were collected for verification purposes and compliance is documented. Participants’ physical activity levels, dietary patterns, changes in medication, and any adverse events experienced within the past two weeks were assessed and recorded by follow-up staff. At 2 and 4 months after the start of the intervention, physical examinations, questionnaires, and biological samples were obtained following the same procedures as at baseline.(3)Final period: 4 months after the intervention, the project ended. Participants who successfully complete the project will receive compensation. At the end of the trial, we unblinded.

### 2.11. Quality Control

(1)Research design stage: review the relevant literature, understand the general situation of relevant research, and design the scheme according to the research purpose; simplify the study design process to minimize unnecessary inquiries and checks; For subjective data collection, standardized scales with high reliability, validity and responsiveness should were used as far as possible. The self-designed questionnaire and the medical records of the subjects were conducted in a small-scale pre-survey among the population before the formal experiment, and then discussed and evaluated with relevant experts and investigators. The final draft was revised several times according to the feedback;(2)Investigator training: the sample collection and data measurement of the subjects was carried out by professional medical workers or trained qualified investigators in the physical examination hospital, and the measurements of each indicator were completed by the same personnel. The investigators were trained and passed the examination;(3)Intervention follow-up stage: during the intervention period, we always communicated with the subjects, and subjects’ medication and lifestyle remained unchanged, we strengthened education to improve the correct understanding and complianceof subjects to the research protocol; simplified study procedures and reduced the number of questions and tests to improve patient patience. Daily supervision was strengthened during the follow-up period, and the compliance of subjects was improved through the daily punch card exchange reward mechanism. We provided a high quality and free medical question answering service for the subjects;(4)Data collection stage: we standardized the basic operation and process of implementation and organized the research subjects to carry out physical examination and questionnaire surveys on time. We strengthened the communication between investigators and research subjects, eliminated the concerns of survey subjects about this study, unified the measurement standards of indicators, reduced information bias, maintained a neutral investigation attitude, and improved the compliance of research subjects.(5)Result analysis stage: Epidata and excel built databases were used to ensure parallel double entry and verification of data. We selected correct statistical analysis methods, and used blind methods for technical personnel, including subject sample processing and statistical analysis, to reduce measurement bias.(6)Data management: according to the original observation records of the subjects, the researcher loaded the data into the physical examination data sheet and various questionnaires in a timely, complete, correct and clear manner. The survey form, which has been reviewed and signed by the Ombudsman, was sent to the research data manager in a timely manner. The corresponding database system was used for two-person and two-machine input, and then the database was compared twice. After all the physical examination data sheets and various questionnaires were entered and verified correctly, the data manager wrote the database inspection report, which included the study completion status (including the list of dropped subjects), inclusion/exclusion criteria, integrity checks, logical consistency checks, outlier data checks, time window checks, drug combination checks, and adverse event checks. After data entry and verification were completed as required, the physical examination data sheets and various questionnaires were filed and stored in numbered order and filled with search directories for reference. Electronic data files, including databases, inspection programs, analysis programs, analysis results, coding and explanatory files, should be classified and stored in different disks or recording media with multiple backups, properly stored to prevent damage. All original files were kept for the period specified accordingly.

## 3. Results

Between 23 August 2022 and 15 October 2022, we recruited 349 elderly people in the Chengdu community who were willing to participate and signed informed consent. Among them, 301 subjects underwent complete geriatric health assessments, including clinical health physical examination, questionnaire surveys, physical function assessment, and anthropometrics as part of the screening process. Blood and stool samples were collected for age-related biomarkers and omics monitoring. Based on inclusion and exclusion criteria, 123 were eligible to participate. The most common reasons for exclusion were not meeting inclusion criteria (*n* = 175) such as having abnormal screening laboratory test values or having cancer or other serious diseases. Of the eligible subjects, 123 subjects were randomized and 61were placed in the nucleotide intervention group and 62 in the placebo control group. However, after randomization, one participant in the nucleotide intervention group decided to withdraw from the trial before starting the intervention, so 122 participants were officially enrolled in the intervention. Figure 4 shows the flow chart of the study samples, including the number of participants who were screened and underwent randomization.

Baseline data on sociodemographic characteristics, co-morbid conditions and health status of the enrolled participants appear in Table 1. All demographic characteristics were well-balanced between groups as expected due to randomization. The mean age was 65.65 ([SD] 2.59) years, 82 (67.21%) were women, and 121 (99.18%) were of Han nationality. Most participants were married (84.43%) and lived with others (91.80%). In terms of education level and monthly disposable income levels, a majority of the subjects received good education and were in good economic conditions. Only 4.10% of them received primary education or below, and only 14.75% had monthly disposable income level less than 2000 yuan. Analysis of smoking and drinking habits showed that only 6.56% and 13.93% of the subjects currently smoke or drink, respectively. The other subjects do not smoke or drink, or had quit smoking and drinking, suggesting that most of them have good living habits.

The baseline results indicate that both the NTs and control groups had similar levels of frail status, physical activity, cognitive function, sleep quality, depression, fatigue, and comorbidities at the start of the trial. Overall, the most common co-morbid conditions were cardiovascular disease (54.10%), renal disease (40.16%), fatty liver (36.89%), chronic respiratory disease (32.79%), hypertension (29.51%), diabetes (13.93%), and dyslipidemia (8.20%). The mean (SD) score of the PASE physical activity scale score was 169.26 (51.93), indicating that most participants were able to engage in moderate physical activity. The SF-12 Physical and Mental Functioning Assessment Scale showed that participants were in good physical and mental health, with a mean (SD) score of 108.00 (7.53). The MoCA Montreal Cognitive Rating Scale score showed an average (SD) score of 21.92 (3.86) out of 30 and only 19.67% of participants had MoCA score ≥ 26, suggesting mild cognitive impairment (MCI) in participants accounting for 65.57%. The PSQI Pittsburgh Sleep Quality Index average (SD) score was 4.89 (3.59) with 40.98% of participants ≥ 5, indicating that nearly half of the participants had clinically significant sleep disturbances. The Kessler Anxiety and Depression Scale average (SD) score was 108.00 (7.53), indicating mild levels of anxiety and depression. The FS-14 Fatigue Scale had an average (SD) score of 4.65 (3.46), with most participants reporting low levels of fatigue. The frailty scale indicated that nearly half of the subjects were in the healthy stage (49.18%) or the pre-frailty stage (46.72%), except for a small number of participants (4.10%) who were in the frailty stage. In conclusion, the Comprehensive Geriatrics Health Assessment Questionnaire indicated that the health of the participants was relatively good overall, with good quality of life and good mental health, but the cognitive and sleep quality remained to be improved.

## 4. Discussions

Quantitative biomarkers of aging are valuable tools for measuring physiological age, assessing the degree of “healthy aging”, and potentially predicting an individual’s health span and lifespan. Given the complexity of the aging process, biomarkers of aging are multilayered and multifaceted. However, most studies so far concentrate on a single measure of biological aging or a single type of measure. Our study comprehensively evaluate the phenotype and molecular biomarkers of aging, placing particular emphasis on evaluating outcomes for markers of aging and possible mechanisms, and comprehensively evaluated the overall health status of older adults.

In our study, the primary outcome measures obtained information from single and multiple systems. For example, telomere length and epigenetic clock methods are cell-level measurements implemented only in a single-tissue blood, while age-related homeostasis, and aging rate measurements take information from multiple systems throughout the body. In addition, we use multi-faceted biomarkers of aging and multi-omics joint analysis to explore the specific mechanisms of targeting the aging process. What is more, the geriatric assessment questionnaire designed for evaluating aging phenotypes, as well as the items included in body composition and physical function measurements in this trial, are reliable and highly efficient. They can systematically, comprehensively, and holistically evaluate the anti-aging effect of nucleotides and their potential to prevent age-related diseases and extend healthy lifespan from a macroscopic.

The TALENTs trial successfully enrolled 122 older adults and completed the baseline assessments. What is more, the intervention and control groups were similar at the beginning of the trial, indicating that the randomization worked well. The results provide valuable information on the demographic and clinical characteristics of the study population and will serve as a reference point for evaluating the effects of NT supplementation on aging outcomes.

Clinical trials tailored to questions of importance to healthy older adults are urgently needed due to increasing numbers of people living into their 80s, 90s, and even 100s. In addition, relatively healthy elderly people have lower risks and higher benefits than those with more diseases, so the general research on anti-aging drugs or food active ingredients would first select relatively healthy elderly people for trial and exploration. Our baseline results of this trial suggest that the study population is a relatively healthy community of elderly aged 60–70 years, with good physical, mental and physical exercise abilities. However, the Montreal Scale indicated that this group of elderly people had mild cognitive impairment (mean 21.92) overall, thus were at increased risk of cognitive decline, which is consistent with the Montreal scores in those aged ≥60 years from a Chinese sub-sample drawn from a population-based study (mean 21.2) [57]. The overall disturbed sleep prevalence of household residents aged 60 and over recruited from a population-based random sample was 33.7%, which was basically similar to the rate of 40.98% in our study population [58]. Together, these data suggest that the baseline status in the current study are fairly representative and comparable to those reported in similar populations. Our intervention still had room for improvement in cognitive level and sleep quality.

This TALENTs trial also has many practical advantages during the implementation process. Firstly, all assessments and the data collection process are entirely in person rather than remote assessment. Secondly, the data collected in the survey are based on previous medical examination reports and available drug packaging, instead of being self- or proxy-reported, and participants were asked to look back in the past within the shortest feasible time to minimize the possibility of recall bias as much as possible. In addition, daily online WeChat online reminders for participants to take the capsules and biweekly home visits greatly reduced dropout rates, and information on daily physical activity, dietary patterns, and medication changes collected during follow-up provided a method for controlling confounding factors.

However, this trial still has certain limitations, such as having a small sample size and a short intervention period as an exploratory trial. Long-term follow-up in future studies will be necessary to assess the lasting effects of nucleotides on aging and longevity outcomes. Additionally, recruitment was performed through online channels, making it difficult for rural populations to participate. People who pay more attention to their health are more likely to learn about this project and participate, resulting in a certain degree of volunteer bias. Despite the limitations of this trial, the results are still of great clinical significance. Our study will reveal the complexity and multi-layered nature of the aging process, providing new directions and perspectives for future anti-aging research. Finally, our trial design and implementation methods also provide references for similar studies.

## 5. Conclusions

In conclusion, this RCT successfully recruited eligible participants and collected baseline data on their sociodemographic characteristics and health status. These data will serve as a reference for evaluating the efficacy and safety of nucleotides as an anti-aging supplement in older adults. The TALENTs trial combines the rigor of traditional double-blind randomized controlled trials with centralized and comprehensive systematic outcome determination, and will provide valuable insights into the anti-aging effects of NTs and contribute to our understanding of their underlying mechanisms. If proven effective, NT intervention may represent a promising strategy for counteracting aging-related decline and improving the quality of life for elderly individuals.

## Figures and Tables

**Figure 1 nutrients-16-01343-f001:**
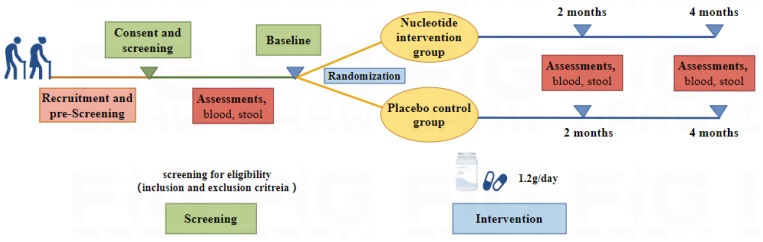
Overview of the TALENTs trial.

**Figure 3 nutrients-16-01343-f003:**
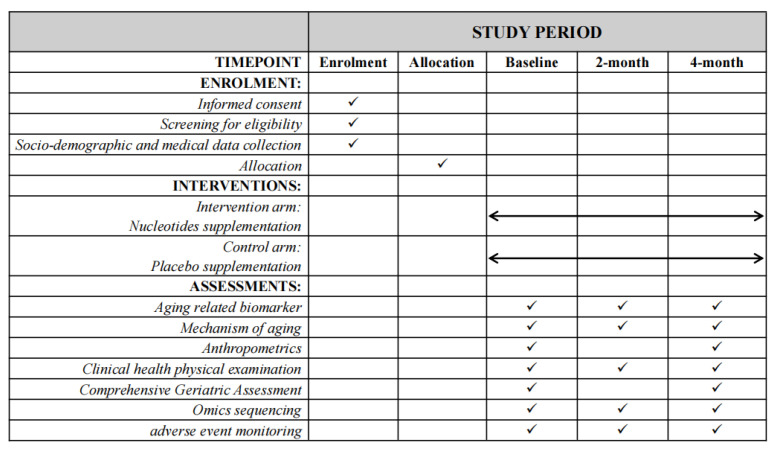
Timeline for scheduled assessments of the TALENTs trial.

**Figure 4 nutrients-16-01343-f004:**
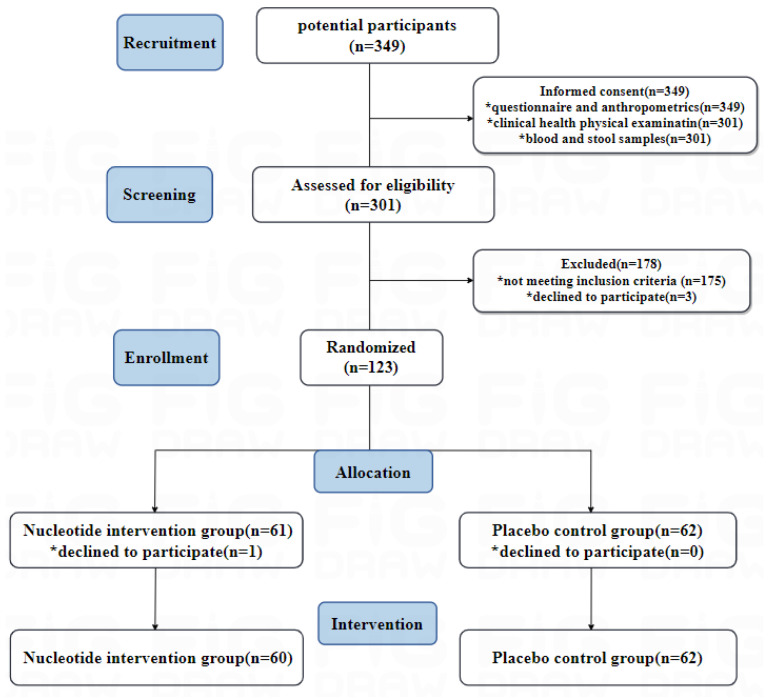
Participant flow of the TALENTs trial. The * represents a change after randomization.

**Table 1 nutrients-16-01343-t001:** Sociodemographic characteristics and health status of TALENTs participants.

	Overall (n = 122)	Intervention Group (n = 60)	Control Group (n = 62)	*p* for Diff
Age, years; mean ± SD	65.65 ± 2.59	65.55 ± 2.63	65.74 ± 2.58	0.685
Female, n (%)	82(67.21)	41(68.33)	41(66.13)	0.950
Nationality, n (%)				
Han nationality	121(99.18)	59(98.33)	62(100)	0.492
Marital status, n (%)				
single	2(1.64)	2(1.64)	0(0)	0.347
married	103(84.43)	50(83.33)	53(85.48)	
divorced/widowed	17(13.93)	8(13.33)	9(6.45)	
Education level, n (%)				
primary school and below	5(4.10)	3(5.00)	2(3.23)	0.116
Junior high school	45(36.89)	25(41.67)	20(32.26)	
High school/technical secondary school	42(34.43)	23(38.33)	19(30.65)	
University/college or above	30(24.59)	9(15.00)	21(33.87)	
Living condition, n (%)				
Live alone	10(8.20)	5(8.33)	5(8.06)	0.957
Non-solitary	112(91.80)	55(91.67)	57(91.94)	
Monthly disposable income, n (%)				
<2000	18(14.75)	9(15.00)	9(14.52)	0.851
2000–3500	47(38.52)	25(41.67)	22(35.48)	
3501–5000	28(22.95)	14(23.33)	14(22.58)	
5001–6500	20(16.39)	9(15.00)	11(17.74)	
>6500	9(7.38)	3(5.00)	6(9.68)	
Smoking status, n (%)				
Never smoked	107(87.70)	49(81.67)	58(93.55)	0.135
Have quit smoking	7(5.74)	5(8.33)	2(3.23)	
smoking	8(6.56)	6(10.00)	2(3.23)	
Alcohol consumption, n (%)				
Never drank alcohol	99(81.15)	48(80.00)	51(82.26)	0.676
Have stopped drinking	6(4.92)	4(6.67)	2(3.23)	
drinking	17(13.93)	8(13.33)	9(14.52)	
Comorbidities, n (%)				
Hypertension	36(29.51)	20(33.33)	16(25.81)	0.362
Dyslipidemia	10(8.20)	2(3.33)	8(12.90)	0.095
Diabetes	17(13.93)	9(15.00)	8(12.90)	0.738
Cardiovascular disease	66(54.10)	37(61.67)	29(46.77)	0.099
Chronic respiratory diseases	40(32.79)	22(36.67)	18(29.03)	0.369
Fatty liver	45(36.89)	22(36.67)	23(37.10)	0.961
Renal disease	49(40.16)	20(33.33)	29(46.77)	0.130
Frailty index, mean ± SD	0.11 ± 0.06	0.11 ± 0.06	0.10 ± 0.06	0.528
Frailty statust, n (%)				
Robust	60(49.18)	27(45.00)	33(53.23)	0.547
Prefrail	57(46.72)	31(51.67)	26(41.94)	
Frail	5(4.10)	2(3.33)	3(4.84)	
PASE, mean ± SD	169.26 ± 51.93	161.98 ± 51.47	176.3 ± 51.82	0.128
SF-12, mean ± SD	108.00 ± 7.53	107.56 ± 7.26	108.42 ± 7.83	0.532
Physical health	52.26 ± 4.38	51.92 ± 3.94	52.58 ± 4.78	0.407
Mental health	55.75 ± 6.3	55.65 ± 6.00	55.84 ± 6.62	0.864
MoCA, mean ± SD	21.92 ± 3.86	21.5 ± 3.87	22.32 ± 3.84	0.241
PSQI, mean ± SD	4.89 ± 3.59	4.57 ± 3.3	5.21 ± 3.85	0.324
Kessler 10 scale, mean ± SD	11.61 ± 2.79	11.37 ± 2.03	11.84 ± 3.37	0.353
FS-14, mean ± SD	4.65 ± 3.46	5.1 ± 3.64	4.21 ± 3.24	0.156
Physical Fatigue	2.46 ± 2.29	2.75 ± 2.42	2.18 ± 2.14	0.168
Mental fatigue	2.19 ± 1.61	2.35 ± 1.69	2.03 ± 1.53	0.277

## Data Availability

The data presented in this study are available on request from the corresponding author. The data are not publicly available due to privacy.

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
