# Peer review of "Targeting Aging and Longevity with Exogenous Nucleotides (TALENTs): Rationale, Design, and Baseline Characteristics from a Randomized Controlled Trial in Older Adults"

_nutrients, 2024, doi:10.3390/nu16091343_

Round 1

Reviewer 1 Report (Previous Reviewer 3)

Comments and Suggestions for Authors

In the article entitled:

Targeting Aging and Longevity with Exogenous Nucleotides (TALENTs): Rationale, design, and baseline characteristics from a randomized controlled trial in Older Adults

authors put their attention toward the aging population problem.

As previously due to the low number of participants I have recommended adding the sentence preliminary studies in the title.

It is well known that after two decades the number of people over 60+ replicated twice in highly developed countries. Therefore, the authors in their study look for the relation between nucleotide supplementation and aging markers like methylation or telomers elongation/shortening. In their randomised double-blind clinical trials, the anthropogenic and biological markers have been taken into consideration. The authors correctly selected mononuclear blood cells as the reference for whole-body conditions (aging). Moreover, the supplementation of monophosphate has been chosen as related to breast milk (neonatal feeding). Additionally, to the chosen method authors mention different aging processes in different tissues.

At this point, it would be nice to mention the biodistribution parameters of nucleotides as a “drug” or diet supplement.

The methodology used for these studies was correctly selected. The article is well written and readable – it was a compact manuscript in the light of the problem and used method.

The authors made some efforts to extend the introduction and discussion part as well as added some figures.

Author Response

Dear reviewer,

  Thank you very much for your comments and suggestions on our paper. In response to your previous question about the small number of participants, we took your suggestion seriously and made the limitations of the study abundantly clear in the discussion. In addition, by synthesizing the sample size of other similar anti-aging studies, we found that our sample size is not particularly small, although it is the first study on nucleotide anti-aging, but at this stage, the direct title is defined as preliminary. In addition, you raise an important question about the biological distribution parameters of nucleotides. We will add relevant content to more fully describe the biological distribution of nucleotides as drugs or dietary supplements. We deeply and sincerely appreciate the reviewer’s endorsement for the content. We made an effort to expand the introduction and discussion section, and added some charts and graphs to improve the readability and understanding of the article. Thank you again for your comments and suggestions, and we hope that these changes will make the article more perfect.

Reviewer 2 Report (Previous Reviewer 2)

Comments and Suggestions for Authors

The authors have adequately addressed/explained my concerns.

Author Response

We sincerely thank the reviewer for your careful review and professional guidance. Your feedback and recognition is a great encouragement to us and greatly promotes the progress of our academic work.

Reviewer 3 Report (New Reviewer)

Comments and Suggestions for Authors

The manuscript titled "Targeting Aging and Longevity with Exogenous Nucleotides (TALENTs)" provides a detailed account of the rationale, design, and baseline characteristics of a randomized controlled trial conducted in older adults. The study focuses on the use of exogenous nucleotides to target aging and longevity, which is a topic of significant interest in the field of aging research and nutrition.

Strengths:

  1. Clear Rationale: The manuscript effectively outlines the rationale behind using exogenous nucleotides to target aging and longevity in older adults.
  2. Robust Design: The randomized controlled trial design is appropriate for addressing the research question and evaluating the efficacy of nucleotides.
  3. Comprehensive Baseline Characteristics: The baseline characteristics provided offer valuable insights into the demographic and clinical profiles of the study participants.
  4. Relevance of Biomarkers: The selection of biomarkers appears appropriate for addressing the research question, which enhances the validity of the upcoming study findings.

Suggestions for Improvement:

  1. Long-Term Follow-Up: Consider including plans for long-term follow-up to assess the sustained effects of nucleotides on aging and longevity outcomes.

Author Response

We are grateful to the reviewer for your careful review and valuable comments. The long-term follow-up recommendations made by the reviewer are very valuable. While current study designs have focused on assessing the effects of nucleotides in the short term, we have recognized the importance of long-term data for understanding the ongoing impact of these interventions. Therefore, we plan to include long-term follow-up in future studies to assess the lasting effects of nucleotides on aging and longevity outcomes. However, the specific implementation will be combined with the approval of ethical changes. Thanks again to the reviewers for their time and effort, your suggestions have been a precious gift to us, which has led us to think more deeply and improve.

This manuscript is a resubmission of an earlier submission. The following is a list of the peer review reports and author responses from that submission.

Round 1

Reviewer 1 Report

Comments and Suggestions for Authors

The manuscript presented by Shuyue Wang and colleagues it is just a protocol for the trial that will be performed in the future. I think is not appropriate for Nutrients.  Gook luck for finding another journal. 

Reviewer 2 Report

Comments and Suggestions for Authors

The study is extensive and comprehensive. 

1. Due to the extensive nature of the study and very long questionnaires for several outcomes, I am concerned about the ability of the participants to adhere to the study. Fallout during the second and third visits is highly likely. Do the authors intend to offer incentives in addition to 500RMB at the end of the four months? Anything that would motivate to stay with the study would be helpful. Simple but useful and eye-catching knickknacks every time the study coordinator visits the participants may keep the participants upbeat. 

2. Can the authors provide details on what type of kits/assays will be used to assess mitochondrial function? Add details in the methods section

3. Immune function: Please include FACS assessment of monocytes and B cells in addition to T cells.   Include IFN gamma and GMCSF in the inflammatory cytokine assessment.

Reviewer 3 Report

Comments and Suggestions for Authors

In the article entitled: Targeting Aging and Longevity with Exogenous Nucleotides in older adults (TALENTs): Trial design and rationale, authors put their attention toward aging population problem. It is well known that after two decades the number of people over 60+ replicated twice in highly developed countries. Therefore, the authors in their study look for the relation between nucleotide supplementation and aging markers like methylation or telomers elongation/shortening. In their randomised double-blind clinical trials, the anthropogenic and biological markers have been taken into consideration. The authors correctly selected mononuclear blood cells as the reference for whole-body conditions (aging). Moreover, the supplementation of monophosphate has been chosen as related to breast milk (neonatal fedding). Additionally, to the chosen method authors mention about different aging processes in different tissues. The methodology used for these studies was correctly selected. However, the investigated group was in my opinion too small therefore I recommend to introduce in the title of the article the preliminary studies phrase. The article is well written and readable – it is a compact manuscript in the light of the problem and used method. I have one critical comment: authors should mention the LADME process of nucleotides as well as the source of NTP in nutrients with their reference ADI.